# Rocks Really Rock: Electronic field trips via Web Google-Earth can generate positive impacts in the attitudes toward Earth sciences, in middle and high school students

Carolina Ortiz-Guerrero[1], Jamie Loizzo[2]

[1]Department of Geological Sciences, University of Florida, Gainesville-Florida, 32611, United States
[2]College of Agricultural and Life Sciences, University of Florida, Gainesville-Florida, 32611, United States

*Correspondence to*: Carolina Ortiz-Guerrero (cog790@gmail.com)

**Abstract.** Earth Sciences (ES) are relevant to society and its relationship to the Earth system. However, ES education, in K-12 environments in the United States, face several challenges including limited exposure to ES, lack of awareness of ES careers, and low ES literacy. International associations have recognized these challenges and recommended that Earth scientists improve the public's perception of the relevance of ES. In recent years, informal science communication/outreach platforms such as the "Streaming Science" model of electronic field trips (EFT), which connect K-12 classrooms with STEM professionals, have gained popularity as an educational technology tool. EFTs are inexpensive, have spatiotemporal benefits, and have proven an effective informal science education pathway for introducing STEM content into formal classrooms to increase positive attitudes and interest in STEM careers. Nevertheless, EFTs in ES for K-12 environments have not been widely disseminated, and their impact in ES education has yet to be studied.

This study presents the creation and implementation of an EFT in geology called "Rocks Really Rock: An Electronic Field Trip across Geological Time." The program was implemented in seven schools in Spring 2022. The EFT was built in web Google Earth and had six stops that featured pre-recorded videos recorded in different locations in Idaho-U.S. The lead presenter/author used multimedia and science-communication strategies such as storytelling to develop and teach concepts related to geologic time, rock formation, and landscape-forming geological process. The content aligned with four specific topics listed in the National Science Foundation's Earth Sciences Literacy Principles and intersected with the Next Generation Science Standards for middle school classrooms.

Participating students (n = 120) completed a post-assessment after the program implementation to evaluate its impact. Results showed the EFT positively impacted students' attitudes toward geology, geology careers, and their perceptions of geology literacy. We identified the three main factors that determined positive attitude change of K-12 students toward ES were: 1) the use of videos and Web Google Earth platform for creating outreach materials for K-12 students, 2) the use of storytelling to craft the content of the EFT, and 3) the asynchronous interactions between teacher-student-scientist. The results indicated a statistically significant positive change in attitudes toward geology, suggesting that participating in the EFT increased students' positive attitudes toward ES. These findings demonstrate the potential of expanding EFT to other ES fields and reaching middle/high school students. We suggest that EFTs are effective outreach tools that can address the challenges in ES

Click here to enter text.

education and can be extended to other ES areas and distributed to students in middle/high schools and
homeschools, to support science educators in ES education.
**1 Introduction**
Earth Sciences (ES) education in U.S. K-12 environments faces multiple challenges such as: 1) low
exposure to ES in the science curricula, 2) low awareness of ES careers, and 3) poor literacy of ES
concepts (Adetunji et al., 2012; Hoisch & Bowie, 2010; LaDue & Clark, 2012). K-12 is used in
reference to the US education system for students from ages 5-18, attending grades between
kindergarten to 12th grade, but this is not solely a US reality. In fact, international associations, ES
educators, and K-12 teachers have recognized these barriers (GSA Position Statement- Promoting Earth
Science Literacy for Public Decision Making, 2013; King, 2013; LaDue & Clark, 2012; Petcovic et al.,
2018), and they have emphasized the need to strengthen K-12 ES education, develop ES-literate
citizens, and advocate for the implementation of informal science-learning strategies (outreach) in K-12
environments. However, there are few studies that have quantitatively assessed the impact of individual
ES' outreach strategies on students.
ES outreach via electronic field trips (EFTs) is a potentially effective way to address some of the
challenges in ES K-12 education. In recent years, the outreach format of EFTs has grown in popularity,
engaging K-12 students and teachers in two-way conversations with subject matter experts. EFT models
such as the Streaming Science model, have proven to be an effective outreach pathway for delivering
science, engineering, technology, and mathematics (STEM) content to formal education environments
such as K-12 classrooms (Adedokun et al., 2011; Beattie et al., 2020; Loizzo et al., 2019). The
adaptability of delivering content in multiple formats (e.g., live-stream or pre-recorded video) and the
ability of EFTs to use science-communication (scicomm) strategies (e.g., digital multimedia,
storytelling) have proven to have a positive impact on students' perceptions and attitudes toward
scientists, science careers, and science overall (Beattie et al., 2020; Dahlstrom, 2014; Loizzo et al.,
2019). These changes in attitudes and perceptions can simultaneously influence interest in related
careers and learning (Lyon et al., 2020; McNeal et al., 2014). Collectively, these findings demonstrate
that the use of EFTs provides a unique opportunity to develop informal ES learning tools and bring
them into formal K-12 education environments.
In the following study, we present the creation, implementation, and evaluation of a pre-recorded EFT
in geology topics created in web Google-Earth called Rocks Really Rock: An Electronic Field Trip
across Geologic Time. The EFT introduced middle-school and high-school students to the concepts of
geologic time, rock formation, and landscape-forming geologic processes. The EFT had six designed
stops shown on a map of the United States. Each stop featured a pre-recorded video of the lead author
who used science communication storytelling strategies to explain geology-related topics that aligned
with four specific topics listed in the Earth Sciences Literacy Principles (ESLP) (Wysession et al.,
2012). The geology topics intersected with the Next Generation Science Standards for middle school
classrooms (NGSS Lead States, 2013). In addition, we examined the implementation of the EFT using a
quantitative design and evaluated the impacts of the program on K-12 school students via a post-

Click here to enter text.

assessment survey in three main areas: a) attitudes toward geology, b) attitudes toward geology careers,
and c) perceptions of geology literacy.
**2. Background Literature**
**2.1 Challenges of ES education and the role of outreach and science communication**
Literacy and awareness of ES topics (e.g. atmospheric sciences, climate sciences, planetary sciences,
environmental sciences, geology, and oceanography) are essential to understanding critical societal
challenges related to the Earth system including climate change, natural resource management, natural
hazards, access to reliable and safe mineral and energy sources, and planetary exploration, among others
(Clary, 2018; Tillinghast et al., 2019; Wysession et al., 2012). Building an ES-literate society depends
on high-quality education, and K-12 school settings have the potential to reinforce positive attitudes
toward ES content and careers and build ES literacy (King, 2013; Levine et al., 2007; St. John et al.,
2021; Tillinghast et al., 2019). However, only a small percentage of students receive formal education
in ES, even in developed countries such as the UK and the United States (Gates & Kalczynski, 2016;
Rogers et al. 2023). In the latter, for example, literacy in ES is particularly low compared to other
scientific disciplines in other countries (Gates & Kalczynski, 2016; Gonzales & Keane, 2011; LaDue &
Clark, 2012; Programme for International Student Assessment & Organisation for Economic Co-
operation and Development, 2019). Furthermore, in countries located in southern Europe and Latin
America, geology courses must share teaching time with other science disciplines, and in countries such
as Australia, geology courses are only available as additional or optional courses (Roca et al., 2020,
Dawborn-Gundlach et al., 2017).
Low exposure to ES content in K-12 environments also impacts the lack of awareness of ES careers
among both students and teachers, and the difficulty students have connecting science classroom
content to career pathways (Brown & Clewell, 1998; Levine et al., 2007; Gonzales & Keane, 2011;
Sherman-Morris et al., 2013; McNeal et al., 2014; Locke et al., 2018, King et al., 2021). Recent
international comparative studies show that three quarters of the countries surveyed recorded that
students have very little, or no careers advise related to ES (King et al., 2021). For example, geology, a
branch of ES, has had the lowest numbers for major recruitment compared to other STEM careers in the
last decades (Levine et al., 2007; Locke et al., 2018), which may be related to an international overall
reduction of university-level ES careers and courses (Geoscience on the chopping block 2021, Rogers et
al 2023).
Several studies suggest that students who choose to study STEM majors generally make the decision
during high school and even earlier (Maltese & Tai, 2011; Tai et al., 2006, Villaseñor et al., 2020).
Thus, growing interest in ES and improving recruitment to ES careers should begin with increased
exposure to engaging STEM content, careers/majors, and raised awareness of future pathways during
middle and high school.  Several strategies have been developed to support formal ES education and
increase awareness and literacy such as integrating ES literacy standards into traditional science courses
(Hanks et al., 2007; Levine et al., 2007; McNeal et al., 2014). For example, in 2011, various Earth
scientists and educators created the Earth Sciences Literacy Principles (ESLP) (Wysession et al., 2012).

Click here to enter text.

The American Geosciences Institute (AGI) has been in charge of disseminating the ESLP, which define the important and essential ES information to be taught, to K-12 ES teachers (Wysession et al., 2012). Furthermore, in the US, the Framework for K-12 Education (National Research Council, 2012), and the subsequent release of the Next Generation Science Standards (NGSS) created a guide for the core ideas and practices that all K-12 students should learn before graduating from high school (NGSS Lead States, 2013). The implementation of these standards introduced a significant amount of ES content into the high school curriculum and increased the emphasis on ES (LaDue & Clark, 2012; Lyon et al., 2020). However, even though the NGSS has placed ES as a core component of the secondary science curriculum, several challenges remain, including the lack of understanding or misunderstanding of ES-related concepts among college-bound students (Pyle et al., 2018), the deficiency of ES instructional resources, the lack of support for school-level ES instruction from the science education community, and the lack of ES-focused teacher training (King, 2013).

Altogether, these challenges in ES education call for a need for new approaches to support the ES K-12 curriculum (King, 2013), such as the reinforcement of students' positive attitudes toward ES through outreach and scicomm. Positive attitudes toward science are a set of affective behaviours such as (1) the manifestation of favourable attitudes toward science and scientists, (2) the enjoyment of science learning experiences, (3) the development of interest in science and science-related activities, and (4) the interest in pursuing a career in science. These behaviours can influence students' interest in science careers and in STEM learning (Fitzakerley et al., 2013; Lyon et al., 2020; McNeal et al., 2014; Osborne et al., 2003). Researchers have commonly measured attitudes toward science using questionnaires with Likert-scale items, which ask students to use a rating scale to indicate a favourable or unfavourable opinion about a statement. The ability to use these responses in statistical analysis has made them a widely used and reliable tool for measuring attitudes toward science topics (Osborne et al., 2003). Moreover, outreach and scicomm have the potential to have a positive impact on the development of positive attitudes toward ES careers and ES literacy. Outreach refers to the activities or processes whose main objective is to promote awareness of STEM in real life, the pursuit of STEM careers, and to motivate non-experts to learn STEM topics (Crawford et al., 2021; Jeffers et al., 2004; Vennix et al., 2017). Outreach programs can take place in person or virtually, and can be structured in a variety of ways, and formats (Crawford et al., 2021). Examples of outreach initiatives include science art installations in nontraditional locations such as public parks (Arcand & Watzke, 2010), the creation of audiovisual material distributed through social media platforms (Gurer et al., 2023), hands-on experiences in nature preserves (Lacey HB, 2016) or museums (Stocklmayer S, 2005), among others. Regardless of their structure or format, outreach activities can use scicomm strategies to achieve these goals, as they they have the potential to increase the comprehension (literacy), interest, and engagement of non-expert science learners (Dahlstrom, 2014), and can be used to increase positive attitudes toward STEM subjects and careers (Burns et al., 2003; Choi et al., 2020; Schmidt & Kelter, 2017). In addition, if the scicomm strategies are aligned with specific learning goals, they can have a positive impact in content area literacy (Hildenbrand GM, 2022).

Click here to enter text.

## 2.2 Electronic Field Trips (EFTs)

Digital outreach strategies such as EFTs have shown the potential to extend scientific research and information about science concepts and careers to a range of formal, informal, and non-formal audiences, allowing viewers to visit virtually any locations around the globe (Beattie et al., 2020; Cassady & Kozlowski, 2008; Evelpidou et al., 2021). For example, The Streaming Science Project is a globally available online outreach platform that includes college-student-created EFTs and other multimedia to introduce audiences to STEM topics and experts. The Streaming Science EFT model (Loizzo et al., 2019) connects science-experts with K-12 students by showcasing live webcasts or pre-recorded videos from various science fields. Using this approach, the Streaming Science EFT model has positively impacted students' perceptions and attitudes about scientists, science careers, and science in general (Barry et al., 2022; Beattie et al., 2020; Loizzo et al., 2019). Wordpress analytics show that more than 137 countries have viewed the Streaming Science overall website since the project began in 2016, and the Rocks Really Rock EFT website had 697 views during 2022-2023 when it was heavily promoted to schools. Science communication materials and outreach programs are publicly available and free as they are often supported through grant funding and faculty and college student research. EFTs can follow different technology formats, from partially to fully immersive augmented reality experiences (usually referred to as virtual field trips), to both pre-recorded and live-streaming video broadcasts, and they can be created using different platforms (e.g., ArcGis Stories, desktop and web-Google-Earth, and virtual reality platforms). Previous studies have shown that students can benefit from virtual field experiences, which have several advantages over in-person field trips, such as: 1) accessibility to learners with all types of abilities and socioeconomic backgrounds, 2) accessibility from any part of the world with an Internet connection, 3) suppression of logistics of in-person field trips such as time, transportation and high costs, 4) availability when sites cannot be visited due to safety conditions, time, weather, or health reasons, and 5) the ability for the audience to move through the content at their own pace (Carabajal et al., 2017; Cliffe, 2017; Evelpidou et al., 2021; Pugsley et al., 2022). EFTs in ES-related topics have been created for formal education at the college level, collecting and processing visual, spatial, and informational data of a geological site of interest with which the user can interact to varying degrees (Barth et al., 2022; Dolphin et al., 2019). Some of these virtual field trips have been created to substitute classic field guides (e.g., Streetcar to Subduction to the San Francisco Bay Area) or to provide remote alternatives to real, in-person field trips in formal ES field education (e.g., virtual field trips during the COVID-19 pandemic) (Bond et al., 2022). These virtual experiences combine digital narratives with geological fieldwork observations, introduce information about a geologic field site, and provide an authentic sense of being at real geological sites (Cliffe, 2017; Dolphin et al., 2019; Granshaw & Duggan-Haas, 2012). Nevertheless, most of these EFTs have been used as an alternative education in ES majors, but they have not been designed with outreach in K-12 environments in mind. Thus, EFTs have the potential to become a widely used outreach strategy in both informal and formal learning environments, following pre-established models for K-12 outreach through EFTs, such as the Streaming Science model (Beattie et al., 2020; Loizzo et al., 2019).

Click here to enter text.

This study examined the development, implementation, and assessment of an EFT called Rocks Really Rock: An Electronic Field Trip across Geologic Time. The EFT followed the Streaming Science EFT model (Loizzo et al., 2019) and a quantitative design to assess the impact of the program on K-12 school students through a post-survey in three main areas: a) attitudes towards geology, b) attitudes towards geology careers, and c) perceptions of geology literacy. The collaboration between scientists and K-12 environments, which this model has successfully tested in several contexts (Aenlle et al., 2022; Barry et al., 2022), provided a platform to positively impact students' attitudes and perceptions toward ES and ES careers using EFTs. In the next section, we describe the development of the EFT and the survey data collection in detail.

## 3. Methods

### 3.1 EFT context and content development

This study developed, implemented, and assessed an EFT called Rocks Really Rock: An Electronic Field Trip across Geologic Time whose target audience was middle and high school students. The EFT consisted of six single-presenter explanatory videos (recorded in Idaho-US in Summer 2021) embedded in a Web Google Earth project, an open-access tool that allows project creators to geotag locations around the Earth and embed multimedia content. Each of the videos was linked to a specific geographical stop with geological significance within the context of the EFT content (Figure 1). The lead author used a storytelling approach to present the content at each of the stops, following a chronological order to tell the story of geological changes on Earth that can be observable in the rocks found in the field. The entire EFT took approximately 40 to 45 minutes and was publicly available online (See supplement link).

The expertise of the subject matter expert (this article's lead author) in the field of geology of Idaho was instrumental in developing the EFT. Ortiz-Guerrero has an academic background in geology and was in the process of finalizing her Ph.D. when she developed the program and assessment. This academic pursuit allowed her to acquire in-depth knowledge and expertise in the subject of the EFT. Furthermore, the EFT content featured her rock research and field sites in Idaho, thus she had familiarity with the regional geological features and their history, which allowed the authors to create a targeted and engaging learning experience for the K-12 students.

The EFT geology content was designed to align with the Next Generation Science Standards (NGSS) learning objectives in the Middle School Earth Sciences (MSESS) disciplinary core ideas, from three subcategories: 1) The History of Planet Earth, 2) Earth's Material and Systems, and 3) Plate Tectonic and Large-Scale System Interactions (National Research Council, 2012; NGSS Lead States, 2013). These NGSS standards also intersect with several of the Big Ideas listed in the National Science Foundation's (NSF) Earth Sciences Literacy Principles (ESLP) (Wysession et al., 2012). Table 1 summarizes the integration of these educational and Big Idea standards, which resulted in the design of the EFT to incorporate four key Big Ideas from the ESLP. The characteristics of each video, the

Click here to enter text.

recording location, and the associated ESLP and NGSS objectives are summarized in Table 2. A unique
sub-website for the EFT was created on the Streaming Science platform, which included a description
of the program, links to a registration form, and the teacher's guide. The teacher's guide was designed
as a stand-alone document that included instructions for K-12 educators to go implement the EFT in
their classrooms.
Storytelling applied to science invites scientists to share their research and learning experiences with
audiences through narrative, making science more accessible and engaging. The overall goal of using
storytelling to explain geology literacy content was to describe selected concepts from the NGSS, in the
context of geochronology and geology careers. Geochronology, referred to by some as "the heart of the
earth sciences" (Harrison et al., 2015), is the discipline that frames the geological events of the earth in a
chronological order. Therefore, by framing the chosen geological concepts within a geochronological
order, the audience was able to follow a narrative arc structure of beginning, middle, and end, allowing
the audience to follow the simple idea of what happened next and learn through the story of Earth's
changes. In summary, the script was constructed to give the audience a reason and a causal connection
between the different geological events at each of the stops, distilling the information to construct a
compelling story, in a non-formal language appropriate to our target audience. In addition to the
geologic story, we introduced the audience to geologic careers by explaining the work of a geologist
using the "AND-BUT-THEREFORE" (ABT) conceptual storytelling structure (Olson, 2015).
The ABT storytelling strategy structures the flow of information by forming a narrative arc in the
audience's mind, avoiding an expository flow of information. In this method, the beginning of the story
presents facts that are connected by "ANDs," which represent an agreement between the facts. In the
middle of the story, the antithesis or problem of the story is introduced by the word "BUT". Finally, the
end of the story follows the antithesis with a solution and is introduced by the word "THEREFORE"
(Olson, 2015). This part gives way to the beginning of the journey, the consequence that leads the
storyteller to the explanation of why we do what we do. To apply this structure in this project, the
ANDs were communicated as geological scientific facts, for example: "The history of the earth is
recorded in the rocks of the earth". The BUT is communicated as an antithesis. For example, "But
geological processes take place on non-human time scales, so we cannot see them. Finally, the
THEREFORE is communicated as a solution: "Therefore, geologists, study the Earth by going into the
field and looking at rocks to study the Earth's history.
**3.2 Research Design**
**3.2.1 Participant Recruitment**
Teacher and student recruitment was conducted after approval by the Institutional Review Board for
Human Subjects Research at the University of Florida. Teachers in K-12 schools in the U.S. were
recruited to participate in the EFT using the following methods: 1) direct email invitation through the
Streaming Science educators' listserv in MailChimp, 2) direct email invitation to educators through the
Scientist in Every Florida School program of the Thompson Earth Systems Institute at the Florida
Museum of Natural History, 3) Streaming Science social media accounts, and 4) word of mouth through
the lead author's personal contacts.

Click here to enter text.

After teachers registered their classrooms for the EFT and indicated their interest in participating in the research, they were emailed a link to the website, teacher's guide, and EFT content. Approved opt-out consent forms were sent home to parents informing them of their child's participation in the EFT and in the anonymous research. Parents who did not want their child to participate had the option of signing and returning the forms to the school. After the forms were returned, teachers implemented the EFT and completed the post-surveys as part of their normal classroom instruction.

### 3.2.2 Survey Design

The student' post-assessment followed a quantitative design to evaluate the impact of the program on K-12 school students through a post-survey in three main areas: a) attitudes toward geology, b) attitude towards geology careers, and c) perceptions of geology literacy. We used a post-retrospective survey design approach which consisted of a questionnaire completed by the students after completing the program. Students were asked to use a rating scale to indicate a favorable or unfavorable opinion about a statement (also known as Likert-scale items). The ability to use these responses in statistical analysis has made them a widely used and reliable tool for measuring attitudes toward science in outreach research (Adedokun et al., 2011; Aenlle et al., 2022; Barry et al., 2022; Lyon et al., 2020; Osborne et al., 2003). In addition, a teacher post-assessment was also implemented to evaluate the teachers' perceptions of the EFT, and to collect suggestions for improving the program. This survey included one open question.
Several questions and statements for the post-retrospective assessment were adapted from previous ES' education studies and EFT studies related to The Streaming Science Project (Adedokun et al., 2011; Lyon et al., 2020; Tillinghast et al., 2019). The student and teacher surveys are available as Supplementary Material (SM1 and SM2). Surveys were implemented using Qualtrics, an online survey platform. The survey link was distributed via email to teachers who had registered to participate. Teachers and students completed the survey electronically or through paper copies that were scanned and sent to the researchers.

### 3.2.3 Data Analysis

Descriptive statistics were used to analyze the quantitative survey data. Paired T-tests with means and p-values were calculated to compare the before and after student responses to the same question. The t-test compares the means between two related groups on the same continuous dependent variable. The greater the magnitude of the t-value, the greater the difference between the means. Conversely, the closer the t-value to 0, the more likely it is that there isn't a significant difference between the means. Each t-value has an associated p-value that indicates the statistical significance of the t, with $p<0.05$ being a statistically significant analysis. The selected valid responses were coded as a data set and analyzed in the SPSS (Statistical Package for the Social Sciences) software to calculate means, standard deviations, t-tests, and p-values.
Several limitations were identified in this study. First, the sample size of participating schools. Although forty-one teachers/classrooms expressed interest in the program, only six classrooms completed the program. Second, some of the students did not complete the entire survey nor did they answer all the

Click here to enter text.

questions, which reduced the amount of useful data. Third, there were problems with the audio quality
in some of the pre-recorded videos in the EFT due to the wind interfering with the microphones during
the field recording portion. The noise, which interfered with the presenters' voice, could have made it
difficult for subjects to understand certain parts of the EFT. However, this difficulty was present in less
than 10% of the materials. Fourth, the limitation of having only one presenter. Although the presenter
had experience with outreach and scicomm, this may have led to audience fatigue. Finally, there was no
detailed demographic assessment which prevented us from distinguishing results between individuals
from different backgrounds.
**4. Results**
The first pilot of the Rocks Really Rock program took place in April and May 2022. Forty-one teachers
initially responded to the Google Form recruitment survey expressing interest in participating in the
program. Six teachers/classrooms participated in the entire program, from EFT presentation to post-
survey distribution and completion. Three classrooms were located in Florida, one classroom in New
York City (homeschool), one classroom in North Dakota, and one classroom in Virginia. Six teachers
answered the whole assessment as reported in Table 7. A total of 120 students participated in the EFT,
and 120 surveys were completed via Qualtrics and paper-copies, which were distributed by teachers
after completion of the EFT to students who did not opt-out of the program.
All the responses were downloaded from Qualtrics and coded as one data set for analysis in SPSS
(Statistical Package for the Social Sciences) software. Surveys with less than 90% of complete
responses were not used for the data analysis. A total of 83 usable student surveys were included in the
data analysis. The survey responses are included as a spreadsheet in Supplementary Materials (SM3).
Figure 3 shows the classroom-grade distribution of participants who completed the post-survey as well
as the gender distribution. Most of the participating students were female. The grade range was 5th-12th
grade. All fifth-grade subjects were from the homeschool participant class. As observed, most of the
participants were middle-school students (6th- 8th grade), and they made up 82% of the sample.
**4.1 Assessing EFT impact on students' attitudes toward geology.**
The first part of the survey attempted to determine how students' attitudes toward geology changed over
the course of the EFT. Students were asked about their attitudes toward geology before and after the
EFT on a scale of 1-6, where 1=unexciting, mundane, and unappealing, and 6 =exciting, fascinating,
and appealing. Table 3 shows the means (M) for the responses to each of the statements for N valid
responses, and the standard deviation (SD) from each mean. The results of the paired t-tests for the
statements are reported for N-t valid responses. Overall, the results show a significant change in
students toward more positive attitudes toward geology after the EFT, as indicated by t-tests and p-
values <0.05. The statement that showed the greatest (and significant) change toward a more positive
attitude was *Geology is appealing/unappealing* (t-test: -5.58, p=0.00). The statement that showed the
least change toward a more positive attitude was *Geology is exciting/unexciting* (t-test: -5.02, p=0.00).

Click here to enter text.

**4.2 Assessing EFT impact on students' attitudes toward geology careers.**

The second part of the survey attempted to determine how the students' attitudes toward geology careers changed due to their participation in the EFT. Students were asked about their attitudes toward geology careers before and after the EFT via a post-retrospective survey using a 5-point Likert-scale with the following range: 1.00=Strongly disagree, 2.00 =somewhat disagree, 3.00=neither agree nor disagree, 4.00 somewhat agree, and 5.00=strongly agree. Table 4 shows the means (M) for the responses to each of the statements for N valid responses, and the standard deviation (SD) from each mean. The results of the paired t-tests for the statements are reported for N-t valid responses, which are the number of answers that can be paired and compared through the test. Statements 2, 3, and 4 showed a statistically significant change in perception, all having p-values <0.05. On the contrary, the t-test for statement 1 is not statistically significant according to the p-value >0.05. The statement that showed the greatest (and significant) change toward a more positive attitude was *Geology is important* (t-test=-5.31, p=0.00). The statement that showed the least change toward a most positive attitude *was Geology is a science* (t-test=-2.47, p=0.02).

**4.3 Assessing impact of the EFT on students' perceptions of geology literacy.**

The third part of the survey attempted to determine how the students' perceptions of geology literacy changed due to the EFT. Students were asked about their attitudes toward geology literacy before and after the EFT using a 5-point Likert-scale with the following range: 1.00=Strongly disagree, 2.00 =somewhat disagree, 3.00=neither agree nor disagree, 4.00 somewhat agree, 5.00=strongly agree Table 5 shows the means (M) for the responses to each of the statements for "N" valid responses. The results of the paired t-tests for the statements are reported for N-t valid responses. All results showed a statistically significant positive change with p-values <0.05. The statement that showed the greatest change was *I have a great deal of knowledge about geology* (t=-8.36, p=0.00).
In addition, students were asked about their knowledge of rocks before and after the EFT on a 5-point Likert-scale with the following range: 1.00=nothing, 2.00=not much, 3.00=a little, 4.00=a lot, and 5.00=everything. Table 6 shows the means (M) for the responses for one question for "N=82" valid responses. The mean score for the question *Before the electronic field trip how much did you know about rocks?* was M=2.93 (SD=0.80), which is between "not much" and "a little," and the mean score for the question *After the electronic field trip, how much do you know about rocks?* was M=3.62 (SD=0.75) which is between "a little" and "a lot." The results of a paired t-test for this statement, for N-t valid responses, showed a positive change in attitude with statistical significance.

**4.4 Assessing teachers' perceptions of the EFT.**

The teachers' survey attempted to determine the teachers' perceptions of the EFT and to know their opinions about the program. Teachers were asked to evaluate their level of agreement or disagreement with thirteen statements using a 5-point Likert-scale with the following range: 1.00=Strongly disagree, 2.00 =somewhat disagree, 3.00=neither agree nor disagree, 4.00 somewhat agree, 5.00=strongly agree Table 7 shows the means (M) for the responses to each of the statements for "N" valid responses. The

Click here to enter text.

teachers' perceptions regarding the students' attitudes was the most positive regarding the statement *The*
*scientist communicated at a level that I understood*. The lowest mean score reported by the teachers was
regarding the statement *The virtual tour inspired my students to want to learn more about careers in*
*geology*. In addition, one open question about opinions and posible improvements was included, and the
answers are reported in Table 8.
**5. Discussion**
According to the Council of Advisors on Science and Technology of the President of the United States,
there will be a shortage of nearly one million STEM professionals in the coming years. Their
projections show that STEM fields will need to increase their recruitment by 34% (Crawford et al.,
2021; Olson & Riordan, 2012). As noted previously, this situation may be more challenging for ES
careers given the lack of exposure/awareness of ES disciplines among K-12 students, in addition to the
low ES literacy of the general population. For this reason, given that high-quality education in K-12
school settings have the potential to reinforce positive attitudes toward STEM content and careers, the
role of these environments is very important in building an ES-literate society and increasing ES career
awareness (Locke et al., 2018). Furthermore, science educators can effectively support these formal
educational settings through outreach activities, which have the potential to increase students' positive
attitudes toward STEM and related careers and increase the motivation to engage in STEM activities
(Vennix et al., 2017, 2018).
The purpose of this study was to determine the impact of an EFT in web Google-Earth on ES topics for
K-12 students. To do so, we built a web Google-Earth EFT using pre-recorded videos called Rocks
Really Rock: An Electronic Field Trip across Geological Time and assessed it with students from seven
middle and high Schools in the United States. Our results showed that EFTs in ES are effective tools
that can be created by Earth scientists to develop outreach projects and support K-12 science educators
to: 1) generate positive attitudes toward the ES, 2) positively impact interest in ES careers, and 3)
reinforce positive perceptions in ES literacy.  In the following section we present our considerations of
this type of EFT and discuss the findings in relation to our research objectives.
**5. 1 Changes in students' attitudes towards Earth sciences using EFT**
The results of this study, in light of the existing literature on STEM and ES outreach, support the
following factors that we believe determine a positive change in K-12 students' attitudes toward ES
using EFTs: 1) the use of pre-recorded videos in the Web Google-Earth platform, 2) the two-way
asynchronous interactions between teacher-student-scientist, and 3) the use of storytelling to design the
content of the EFT. Here, we lay out the main considerations that led us to propose these factors.
**5. 1.1 Use of pre-recorded videos in Web Google-Earth.**
There are several advantages (for both creators and users) of Web Google-Earth as a platform for
creating virtual field trips in the ES, such as: the effective and user-friendly format and interface of the
platform, the easy way to distribute via direct web link, the ability to geotag the different field trip stops
in one single project, the 3D view navigation of the locations providing opportunities for independent
exploration, among others (Barth et al., 2022; Evelpidou et al., 2021; Mahan et al., 2021; Wyatt &
Werner, 2019). In addition, EFTs through Web Google-Earth do not limit the experience to the
geotagged locations, but also allow the creator to include links to supporting materials (e.g., links to
publications, maps, field guides, among others) and display multimedia content (photos, videos, satellite
images, slides) that allow the user to further explore the studied area (Evelpidou et al., 2021).
One of the more powerful outreach benefits of Web Google Earth is the use of multimedia, particularly
video. Several studies have shown that multimedia in both science education and outreach can present
science materials effectively, efficiently, and more interestingly, which helps students engage with
science content and achieve learning outcomes (Morris & Lambe, 2017; Syawaludin et al., 2019; Wang
et al., 2022). For example, pre-recorded videos in ES are known to increase interest in STEM because
they provide a way to present content knowledge to the public using images, text, multimedia, etc.,
which can also create a different pedagogical experience (Wang et al., 2022). We suggest that ES
outreach programs through Web Google Earth can benefit from the possibility of combining two tools:
pre-recorded ES videos and geotagged locations. This allows students to follow the presenter's
explanations, experience the presenter's field observations at each site, and explore the geotagged
locations where the videos were filmed. The pre-recorded videos also allowed us to embed explanatory
graphics and videos from other creators. Our videos can be easily found by other ES educators on
YouTube and can be used in various teaching and learning environments, as accessible support
materials for other ES educators around the world (Maynard, 2021; Welbourne & Grant, 2016).
**5.1.2 Asynchronous interactions between teacher-student-scientist.**
The benefits of interactions between students, teachers, and scientists have been previously evaluated
and found to be an essential part of science outreach by positively changing students' perceptions of
science and science-related careers (Barry et al., 2022; Painter et al., 2006, Rogers et al., 2023).
International organizations science organizations, researchers and K-12 science educators across the
globe believe that there is a need for scientists to be involved in science education (GSA Position
Statement- Promoting Earth Science Literacy for Public Decision Making, 2013; King, 2013; Levine et
al., 2007). Currently, several ES K-12 outreach strategies for students and teachers focus on in-person
visits from professional scientists, visits to science fairs, visits to science museums, and field trips
(Abramowitz et al., 2021; Onstad, 2021; Tillinghast et al., 2019). However, many of these outreach
strategies have limitations, including lack of funding for in-person visits, time-consuming
transportation, or accessibility.
Our results showed that outreach through EFTs in Web Google Earth is an asynchronous alternative for
interactive learning experiences in formal educational environments (K-12 classrooms). This mode of
EFT has the potential to create positive attitudes toward ES and ES careers, similar to previous
synchronous interactions through EFTs via the Streaming Science model (Barry et al., 2022; Loizzo et
al., 2019). Because the core of the EFT activity is asynchronous, it has the advantage of being used
multiple times by students and teachers after the class activity, and it allows the teacher to view it prior
to the class activity. This is supported also by one of the responses to the teachers' survey; "The EFT
went well because we could complete it at our pace. I could go to the places on the map that my
students wanted to look at". Additionally, the asynchronous, pre-recorded nature of the EFT reduces
barriers for students and teachers who may face barriers to accessing field-based outreach events due to
financial limitations or physical disabilities (among others), allowing for inclusive participation in
outreach activities.
**5.1.3 The use of storytelling to craft the content of the EFT.**
Several studies have highlighted that ES is a challenging set of sciences to communicate to non-expert
audiences (Scherer et al., 2017; Sell et al., 2006). Wang et al. (2022) proposed three categories to
explain the challenges of communicating ES topics: 1) Earth processes operate at unobservable
locations and nonhuman "deep timescales," 2) ES information is more relevant to some locations than
others, and 3) ES topics involve complex and dynamic systems. Therefore, regardless of the accuracy of
the content of an ES outreach strategy, it may not always be effective in positively impacting the
learning experience of non-expert audiences or in engaging them with scientific content. However, there
are several science communication tools that geoscientists can use to effectively communicate ES to the
public, such as science storytelling (McNeal et al., 2014; Stewart & Hurth, 2021), and within
storytelling several tools that may help science stories to engage the targeted audience, such as the ABT
structure (Olson R, 2015).
Our research supports previous research that suggests that science communication through storytelling
is an effective strategy for achieving positive impacts through ES outreach initiatives (Dahlstrom, 2014;
Joubert et al., 2019; Martinez-Conde & Macknik, 2017, Rogers et al., 2023). In this study, the presenter
used a storytelling approach using a chronological narrative to present facts and evidence about Earth's
history, allowing students to go through the science content as if they were being told the story of Earth
through time. In addition, applying the "ABT" structure to showcase geology careers, provided a
framework to justify the role of geologists in understanding the history of Earth. Our results show
overall that the content of our pre-recorded videos was effective in promoting interest with the ES and
ES careers, suggesting that storytelling may contribute significantly when developing asynchronous
science outreach material for K-12 students.
**5.2 Addressing the challenges in ES education and ES careers through outreach.**
The study discussed in this article focused on the evaluation of attitudes toward geology and Earth
sciences (ES) education using an Earth Field Trip (EFT) intervention. The results of t-tests indicated a
statistically significant positive change in attitudes toward geology, suggesting that participating in the
EFT increased students' positive attitudes toward ES. These findings demonstrate the potential of
expanding EFT to other ES fields and reaching middle/high school students. These findings align with
previous research on STEM education and outreach, emphasizing the significance of positive attitudes
and well-informed perceptions in fostering interest in ES learning and pursuit of ES careers. In the
following section we discuss the following topics: 1) the role of EFTs in students' attitudes toward
Earth sciences, and 2) The role of EFT in Earth sciences in the perception of ES literacy.

Click here to enter text.

### 5.2.1 The role of EFTs in students' attitudes toward Earth sciences.

The t-tests made for the statements regarding attitudes toward geology (e.g., *Geology is unexciting/exciting, Geology is mundane/fascinating, and Geology is appealing/unappealing*) showed a statistically significant positive change, indicating that attitudes toward ES increased after students participated in EFT. These findings demonstrate the feasibility of expanding EFT to other ES fields (not just geology) and to middle/high school (and home) students. Thus, EFT may help science educators change negative or neutral attitudes toward ES to positive attitudes. In addition, EFT may address teacher unpreparedness for ES content and the paucity of available interactive ES instructional resources that prevent and limit ES instruction in various K-12 settings (King, 2013).
Based on our findings, the lack of awareness of ES may not be as much of a challenge for ES education (as reported in the literature) as the lack of enthusiasm for ES among K-12 students. Our results showed that there was no statistically significant change when we measured awareness, as most students were aware of geology as a science and where geologists might work before the EFT. However, the t-tests related to the statements measuring attitudes toward geology and geology careers all showed significant positive results.
Research has shown that students considering geology careers do so as early as middle school (Lyon et al., 2020). Thus, the use of EFT in this stage can become a powerful intervention strategy to influence ES career choices in a positive way. Based on our findings, there was a significant positive change after following the EFT, on attitudinal statements about geology careers in both the student and the teachers survey (e.g. *A job as a geologist would be interesting, I would consider geology as a major, geology is important,* and *The virtual tour inspired my students to want to learn more about careers in geology*.) Therefore, such EFTs can combine K-12 ES topics (linking learning goals to ESLPs or NGSS) with real-world career scenarios to increase students' interest in ES careers. These EFTs can address students' difficulties connecting science content to career pathways, as well as educators' lack of knowledge about realistic role models in these careers (Jahn & Myers, 2015; Levine et al., 2007; Lyon et al., 2020; McNeal et al., 2014; Petcovic et al., 2018). We recognize that the implementation of this EFT in the science classroom did not necessarily indicate successful recruitment of students into an ES major, but the data demonstrated that the EFT was successful in positively impacting students' thoughts about choosing a geology major.
All findings discussed in this article support previous STEM education and outreach research in ES and other STEM fields. Prior research has shown that an EFT as outreach strategy can support STEM education by fostering positive attitudes toward science, which tends to encourage youth to pursue STEM careers and build a skilled STEM workforce (Barry et al., 2022; Loizzo et al., 2019). Similarly, several studies in ES education remind us that positive attitudes and well-informed perceptions about the field of geology and other ES fields influence middle and high school students' interest in ES learning and desire to pursue ES careers (Kurtis, Kimberly, 2009; Lyon et al., 2020; McNeal et al., 2014).

Click here to enter text.

### 5.2.2 The role of EFT in Earth sciences in the perception of ES literacy.

Our study found that an EFT built in web Google Earth covering ES topics had a positive impact on students' perceptions of geology literacy and their interest in learning geology topics. After students completed the retrospective self-assessment of their knowledge of ES, there was a statistically significant positive difference in the pre-post statements. The change in the statement *I have a great deal of knowledge about geology* indicated that the EFT had a positive impact on the students' perception of their knowledge of ES, and that this perception improved. Similarly, the change in the statement *I would like to learn more about geology* showed that students had an increased desire to learn and an increased interest in geology after the EFT.

Our study contrasts to other studies that have assessed students' perceptions and interest in ES literacy by exposing K-12 students to ES content but have not necessarily obtained positive attitudinal changes after the programs. For example, Lyon et al. (2020) used the statement *I would like to learn more about geology* in an attitudinal survey program in ninth graders who had been exposed to a Geosciences course with content aligned to the NGSS. Their data showed a decrease in interest in geology on the post-survey after had taken the course. The authors considered that one of the main challenges may have been in "translating material covered in class into something they (the students) value" (Lyon et al., 2020). The difference in results between an ES course and an ES outreach program such as our EFT supports our previously mentioned premise about how ES topics are communicated (using storytelling and multimedia) and supports the idea that in K-12 settings, ES outreach using multimedia and science communication tools may be more effective in generating positive attitudes toward geology than exposing students to ES courses.

Although our study focused on the U.S. education system, several challenges of ES education and careers are shared by several other countries, as mentioned above. Thus, this strategy has the potential to be implemented globally and to complement or cover gaps in the ES curriculum at the primary and secondary levels and to work towards improving awareness of ES careers (King et al., 2021). For example, in countries such as Chile, researchers have found that the ES K-12 school curriculum is not relevant and have therefore called for the implementation of educational experiences related to ES (Villaseñor et al., 2020), for which EFTs may also work.

### 5.3 Recommendations: How can the implementation of Earth Sciences electronic field trips be improved?

Based on this pilot study using web Google-Earth for ES outreach in K-12 environments we consider a number of recommendations for EFT creators, users, as well as for further research. Creators, especially scientists with no experience multimedia creation, may find it useful to allocate funding to work with expert multimedia editors to fund the participation of other subject-matter-experts during the video recordings, to integrate dialogue and conversation among the presenters, as noted by one of the responses to the teachers' survey. Funding may also be allocated to improve the video and audio quality of the delivered content. In addition, more content can be added to each site between longer-form videos if there is an opportunity to explore more sites in the area. By making more content available at multiple geo-tagged locations, students and teachers will be able to engage with the application in a more interactive way.

Click here to enter text.

The EFT is adaptable to many ways of class instruction, whether it is more individual or group-focused.
We suggest that the teachers first go through the Google Earth web program on their own before
presenting it in their classrooms, and if deemed appropriate, design exercises using the concepts learned
in the EFT that can complement the activity before, during, or after the EFT is presented to students,
similarly to this teacher's idea: "When we visit again, I will create a work sheet for the students to take
notes during the presentation and another to sum up what they have learned." Teachers can also network
with the creators and participate in annual research to assess the impact of these EFTs at different K-12
levels to determine which groups of students are more or less impacted. These strategies, altogether,
may potentially reduce the impact of our previously-identified limitations to the outreach program, such
as the technical difficulties of recording videos in the outdoors, or the audience fatigue that may be
caused by single presenter videos, both included on the recommendations teachers gave to this first pilot
program (Table 8).
**6. Conclusions**
Earth Sciences are relevant to society and its relationship to the Earth system. However, ES education in
U.S. K-12 environments faces multiple challenges such as 1) limited exposure to ES, 2) lack of
awareness of ES careers, and 3) low ES literacy. Interactions between science educators, students, and
scientists are an essential part of science outreach. Previous studies have shown that successful outreach
programs leading to positive attitudinal changes toward STEM in students can help students understand
how science can explain the natural world around them.
This study found that outreach through EFTs in Web Google Earth is an asynchronous alternative to
synchronous interactive learning experiences in formal education environments (K-12 classrooms.) Our
study showed that web Google-Earth EFTs have the potential to increase positive attitudes toward ES
(specifically geology), interest in ES careers, and perceptions of ES-literacy, providing several
advantages for ES K-12 outreach. The use of EFT for ES outreach presents a unique opportunity for
Earth Scientists located not only in the United States but anywhere in the globe, to network with K-12
educators and address these challenges, creating interactions between scientists and K-12 classrooms.
Our findings indicated that one of the major problems in ES education is not a lack of awareness but a
lack of excitement among K-12 students about ES topics, and therefore scicomm tools such as
storytelling and use of multimedia in platforms such as web Google Earth, provide an effective strategy
for creating outreach content that generates engagement with science topics and increases positive
attitudes toward science.

Click here to enter text.

Figures:

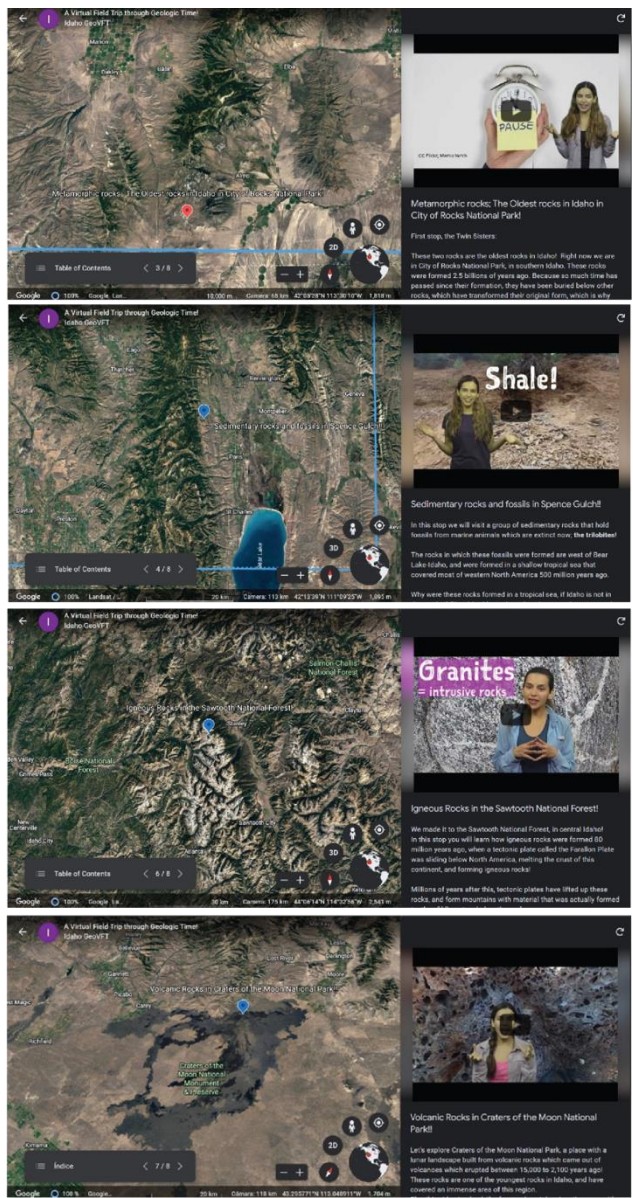

Figure 1. Screenshots from the EFT "Rocks Really Rock, and EFT Across Geological Time". Adapted
from: https://earth.google.com/earth/d/1btfkYpOkcsqQktfky-t0pYJLT1e2lJSP?usp=sharing © Google
Earth 2023. Recovered: September 19, 2023

Click here to enter text.

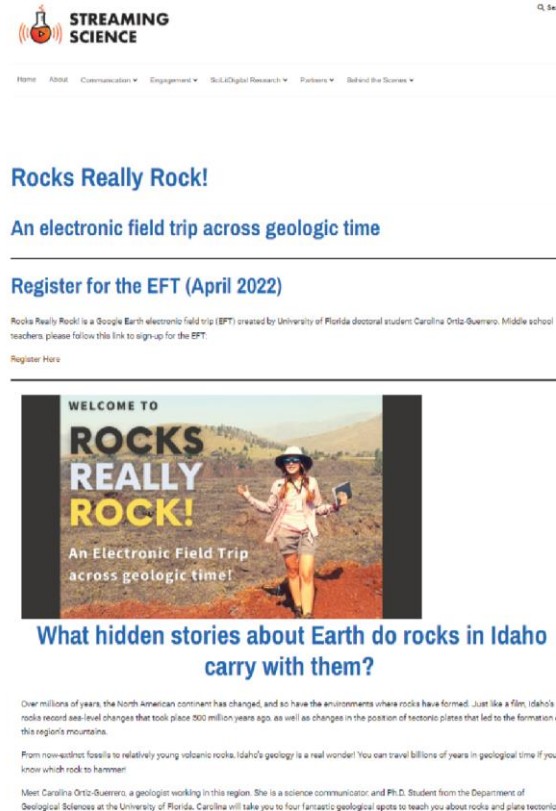

Figure 2. Screenshot from Streaming Science web page for "Rocks Really Rock EFT". Adapted from:
https://streamingscience.com/rocks-really-rock-an-electronic-field-trip-across-geologic-time/
Recovered: September 19, 2023

Click here to enter text.


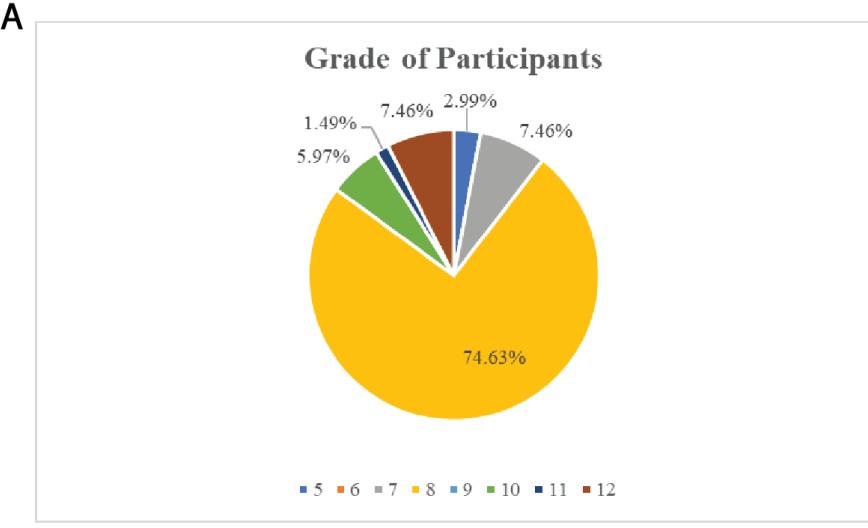

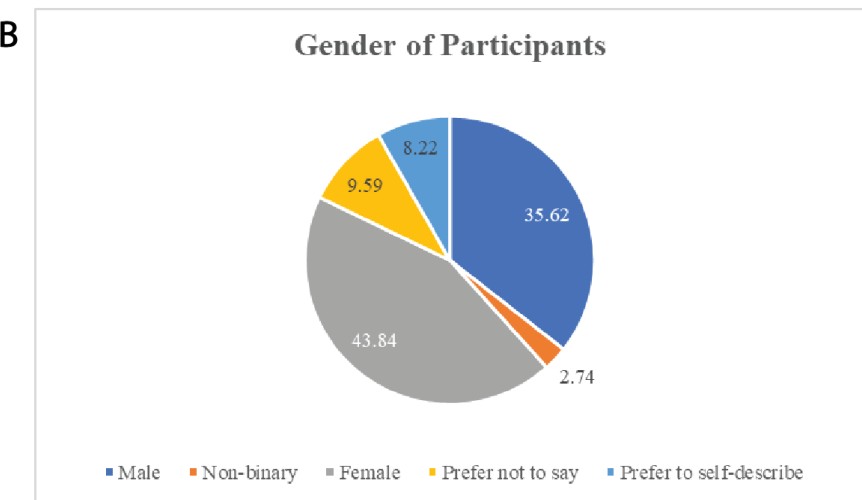

Figure 3. A) Grade distribution from participant students. B) Gender distribution from participant
students.

Click here to enter text.

Tables
Table 1. List of *Earth Sciences Literacy Principles* (ESLP) and *Next Generation Science Standards*
(NGSS) used for content literacy in "Rocks Really Rock" EFT

| ESLP | Middle School Earth Sciences (MS-ESS) NGSS standards used in content creation |
| --- | --- |
| Big Idea 2 (Earth is 4.6 billion years old) | MS-ESS1.C - The History of Planet Earth. |
| | MS-ESS2.A - Earth's Material and Systems |
| | MS-ESS2.B - Plate Tectonic and Large-Scale System Interactions |
| Big Idea 3 (Earth is a complex system of interacting rock, water, air, and life). | MS- ESS1.C - The History of Planet Earth. |
| | MS-ESS2.A - Earth's Material and Systems |
| | MS-ESS2.B - Plate Tectonic and Large-Scale System Interactions |
| Big Idea 4 (Earth is continuously changing) | MS- ESS1.C - The History of Planet Earth. |
| | MS-ESS2.B - Plate Tectonic and Large-Scale System Interactions |
| Big Idea 6 (Life evolves on a dynamic Earth and continuously modifies Earth). | MS- ESS1.C - The History of Planet Earth. |


Click here to enter text.

Table 2. Structure of "Rocks Really Rock" EFT

| Video/ Duration (mins/secs) | Recording Location | Covered Topics, Earth Science Literacy Principle (ESLP), and Next Generation Science Standard (NGSS) | Learning Objectives |
|---|---|---|---|
| 1. Intro (2m 24s) | Studio | This module is an introduction into the program and to the concepts of geologic time, and plate tectonics.<br><br>ESLP=Big Idea 2 (Earth is 4.6 billion years old), and Big Idea 3 (Earth is a complex system of interacting rock, water, air, and life).<br><br>NGSS=MS- ESS1.C, The History of Planet Earth. | 1. Recall what is the geologic timescale. |


Click here to enter text.

Table 2. Continued

| Video/ Duration (mins/secs) | Recording Location | Covered Topics, Earth Science Literacy Principle (ESLP), and Next Generation Science Standard (NGSS) | Learning Objectives |
| --- | --- | --- | --- |
| 2. Stop 1 "City of Rocks, Looking for the oldest rocks in Idaho" (5m 29s) | Twin Sisters rocks at City of Rocks National Park (Idaho-US) +Studio | This module covers three different topics: 1) The age of the oldest rocks in Idaho, 2) The differences between today's Earth and Earth 2-billion years ago, and 3) the concept of metamorphism. ESLP=Big Idea 2 (Earth is 4.6 billion years old), and Big Idea 4 (Earth is continuously changing). NGSS= MS- ESS1.C, The History of Planet Earth., and MS-ESS2.A-Earth's Material and Systems | 1.Recall what is a metamorphic rock. 2.Recall how old are the oldest rocks in Idaho. |


Click here to enter text.

Table 2. Continued

| Video/ Duration (mins/secs) | Recording Location | Covered Topics, Earth Science Literacy Principle (ESLP), and Next Generation Science Standard (NGSS) | Learning Objectives |
|---|---|---|---|
| 3. Stop 2 "Cambrian Fossils". (5m 21s) | Spence Gulch (Idaho-US) +Studio | This module covers four different topics: 1) Changes in Earth from 2000-500 Ma, 2) The Cambrian Earth and the Cambrian explosion 3) Formation of sedimentary rocks, and 4) Formation of fossils, and ichno-fossils.<br><br>ESLP=Big Idea 2 (Earth is 4.6 billion years old), Big Idea 4 (Earth is continuously changing), and Big Idea 6: Life evolves on a dynamic Earth and continuously modifies Earth.<br><br>NGSS= MS- ESS1.C, The History of Planet Earth., and MS-ESS2.A-Earth's Material and Systems. and MS-ESS2.B Plate Tectonic and Large-Scale System Interactions. | 1. Recall what is a sedimentary rock 2. Recall what is a fossil, and what is a trilobite. 3. Recall what was the Cambrian explosion. |


Click here to enter text.

Table 2. Continued

| Video/ Duration (mins/secs) | Recording Location | Covered Topics, Earth Science Literacy Principle (ESLP), and Next Generation Science Standard (NGSS) | Learning Objectives |
|---|---|---|---|
| 4. Subduction Zone and Plate Tectonics (2m57s) | Studio | This module explains the formation of subduction zones, and the occurrence of a subduction zone in the Cretaceous in western North America.<br><br>ESLP=Big Idea 2 (Earth is 4.6 billion years old), and Big Idea 4 (Earth is continuously changing)<br><br>NGSS= MS- ESS1.C, The History of Planet Earth., and MS-ESS2.A-Earth's Material and Systems. and MS-ESS2.B Plate Tectonic and Large-Scale System Interactions | 1.Recall the effect of the movement of plate tectonics, in changing the shape of continents. |


Click here to enter text.

Table 2. Continued

| Video/ Duration (mins/secs) | Recording Location | Covered Topics, Earth Science Literacy Principle (ESLP), and Next Generation Science Standard (NGSS) | Learning Objectives |
|---|---|---|---|
| 5. Stop 3 "Igneous Rocks in the Sawtooth Moutain" (6m13s) | Sawtooth Lake at the Sawtooth National Forest (Idaho-US) +Studio | This module covers three topics: 1) Plate tectonics 80 million years ago in The Cretaceous, 2) Formation of igneous rocks in subduction zones, 3) Minerals forming granitic rocks, and 4) geology methods for outcrop rock observation.<br><br>ESLP=Big Idea 2 (Earth is 4.6 billion years old), and Big Idea 4 (Earth is continuously changing)<br><br>NGSS= MS- ESS1.C, The History of Planet Earth., and MS-ESS2.A-Earth's Material and Systems. and MS-ESS2.B Plate Tectonic and Large-Scale System Interactions | 1.Recall what is a subduction zone, and the effects on mountain formation. 2. Recall what an igneous rock is. |
| 6. Stop 4 "Origin of volcanic rocks" (6m14s) | Craters of the Moon National Park (Idaho-US) +Studio. | This module covers two topics: 1) Formation of volcanic extrusive rocks, and 2) Formation of lava tubes.<br><br>ESLP= Big Idea 4 (Earth is continuously changing). NGSS= MS- ESS1.C, The History of Planet Earth., | 1.Recall what type of rock a basalt is. 2.Recall what are lava tubes. |


Click here to enter text.

Table 3. Survey results about students' attitudes towards geology before and after EFT. The table
presents the Mean score for two statements with the following ranking scale: 1 = unexciting, mundane,
unappealing /// 6=exciting, fascinating, appealing. N participants were surveilled, and N-t valid answers
were taken into account to calculate the T-test value and its corresponding P-value.

| Statements BEFORE the 'Rocks really rock' electronic field trip, I thought Geology was | Mean score. (Standard Deviation) | Statements AFTER the 'Rocks really rock' electronic field trip, I now think Geology is | Mean score. (Standard Deviation) | N | T-test before & after | P-value (Sig. 2-tailed) | N-t |
|---|---|---|---|---|---|---|---|
| unexciting-exciting | 2.99 (1.27) | unexciting-exciting | 3.72 (1.36) | 83 | -5.02 | 0.000 | 82 |
| mundane-fascinating | 3.33 (1.35) | mundane-fascinating | 4.00 (1.36) | 83 | -5.08 | 0.000 | 82 |
| unappealing-appealing | 3.23 (1.43) | unappealing-appealing | 4.01 (1.38) | 83 | -5.58 | 0.000 | 82 |

Table 4. Survey results about students' attitudes about geology careers. The table presents the Mean
score for two statements with the following ranking scale cale: 1 = Strongly disagree, 2=Somewhat
disagree, 3=Neither agree nor disagree, 4=Somewhat agree, 5=Strongly Agree. N participants were
surveilled, and N-t valid answers were considered to calculate the T-test value and its corresponding P-
value

| Statements | Mean score. (Standard Deviation) | | N | T before & after | P-value (Sig. 2-tailed) | N-t |
|---|---|---|---|---|---|---|
| | BEFORE participating in the Rocks really Rock EFT, I thought | AFTER participating in the Rocks really Rock EFT, I now think | | | | |
| | 4.49 | 4.61 | | -1.32 | 0.19 | |

Click here to enter text.

| Statement | Mean before (SD) | Mean after (SD) | N | T-test | P-value | N-t |
|---|---|---|---|---|---|---|
| Geologists can work outdoors. | (0.79) | (0.71) | 83 | | | 82 |
| Geology is a science. | 4.26 (0.89) | 4.49 (0.77) | 82 | -2.47 | 0.02 | 81 |
| Geology is important. | 3.71 (1.02) | 4.23 (0.85) | 83 | -5.31 | 0.00 | 82 |
| A job as a geologist would be interesting. | 2.66 (1.07) | 3.12 (1.14) | 82 | -3.93 | 0.00 | 81 |
| I would consider geology as a major | 2.09 (1.06) | 2.43 (1.17) | 81 | -3.64 | 0.00 | 80 |

Table 5. Survey results about students' perceived literacy in geology Pt1. The table presents the Mean
score for two statements with the following ranking scale: 1 = Strongly disagree, 2=Somewhat disagree,
3= Neither agree nor disagree, 4= Somewhat agree,5= Strongly agree. N participants were surveilled,
and N-t valid answers were considered to calculate the T-test value and its corresponding P-value.

| Statements BEFORE participating in the Rocks really Rock EFT, I thought | Mean score. (Standard Deviation) | Statements AFTER the 'Rocks really rock' electronic field trip, I now think Geology is | Mean score. (Standard Deviation) | N | T-test before & after | P-value (Sig. 2-tailed) | N-t |
|---|---|---|---|---|---|---|---|
| | | | | | | | |

Click here to enter text.

| | | | | | | | |
|---|---|---|---|---|---|---|---|
| *I have a great deal of knowledge about geology.* | 2.66 (1.00) | *I have a great deal of knowledge about geology.* | 3.46 (0.89) | 83 | -8.36 | 0.00 | 82 |
| *I would like to learn more about geology.* | 2.84 (1.07) | *I would like to learn more about geology.* | 3.40 (1.20) | 82 | -5.54 | 0.00 | 81 |


Click here to enter text.

Table 6. Survey results about students' attitudes about perceived literacy in geology before and after the
EFT Pt2. The table presents the Mean score for two statements with the following ranking scale: 1=
Nothing, 2= Not much,3=A little, 4=A lot, 5=Everything. N participants were surveilled, and N-t valid
answers were considered to calculate the T-test value and its corresponding P-value.

|  | Mean score. (Standard Deviation) Students' attitudes | N |
|---|---|---|
| *Before the Electronic Field Trip how much did you know about rocks?* | 2.92 (0.80) | 82 |
| *After the Electronic Field Trip how much do you know about rocks?* | 3.62 (0.75) | 82 |
| T-test | -9.53 | |
| P-value | 0.00 | |
| N-t | 81 | |


Click here to enter text.

Table 7. Survey results about teachers' perceptions of the EFT. Scale: 1 = Strongly disagree,
2=Somewhat disagree, 3=Neither agree nor disagree, 4=Somewhat agree, 5=Strongly Agree

| Statements | Mean score. (Standard Deviation) | N |
|---|---|---|
| *The topic was interesting.* | 4.83 (0.41) | 6 |
| *The scientist was interesting.* | 4.83 (0.41) | 6 |
| *The scientist talked about something I did not already know.* | 4.33 (0.82) | 6 |
| *The scientist communicated at a level that I understood.* | 5 | 6 |
| *The scientist was knowledgeable about the topic.* | 4.83 (0.41) | 6 |
| *The scientist gave an interesting demonstration to explain the origin of rocks.* | 4.33 (1.21) | 6 |
| *It is important that we learn about Earth's history.* | 4.83 (0.41) | 6 |
| *I learned about careers in geology from the scientist.* | 4.17 (0.75) | 6 |
| *I would recommend this electronic field trip to other classes.* | 4.66 (0.52) | 6 |
| *My students were engaged with the virtual tour.* | 3.83 | 6 |

Click here to enter text.

|  | | |
|---|---|---|
|  | (0.98) | |
| *The virtual tour inspired my students to ask questions about geology.* | 3.83 (0.41) | 6 |
| *The virtual tour inspired my students to want to learn more about careers in geology.* | 3.17 (0.75) | 6 |
| *The electronic field trip was easy to hear.* | 4.33 (1.21) | 6 |


Click here to enter text.


Table 8. Survey results about teachers' opinions of the EFT

| Respondent | Survey indication: *Please leave a comment about what went well and didn't go well by using the EFT. If you have any suggestions for improving the program, write them below.* |
|---|---|
| 1 | "It is best to share the EFT as whole class.  Using ipads or chromebooks has issues with school wifi.  It would be neat to have a live virtual EFT." |
| 2 | "They EFT went well because we could complete it at our pace. I could go to the places on the map that my students wanted to look at." |
| 3 | "I enjoyed the multiple sites. The camera and mic quality were great. The conversation was a little stiff and could use a second scientist to conversate with." |
| 4 | "No problems with using the link or the videos. The sound quality when outdoors was sometimes a little difficult to hear/understand due to the wind. The indoor recording had echo. I presented the EFT on a SmartBoard so all students could watch. [..]" |
| 5 | "The students liked seeing the rocks in their natural habitat. When we visit again, I will create a work sheet for the students to take notes during the presentation and another to sum up what they have learned. A link to more information would be helpful too. Some of the students commented that the volume changed and that you could hear the wind. A fluffy microphone might help with that. Overall, we liked the trip and I plan on using it again in the future." |
| 6 | "Using EFT was very easy and instructions were clear in how to navigate through it and what to do to prepare and send opt-out options for parents. Some of the information was hard to hear with the way some of the videos were recorded." |



Click here to enter text.

## Video Supplement

The following link contains the public web-address to the electronic field trip "Rocks Really Rock"
which take viewers to the web-Google Earth application
https://earth.google.com/earth/d/1btfkYpOkcsqQktfky-t0pYJLT1e2lJSP?usp=sharing

## Author contribution

COG and JL: concept, data collection, research, writing, edition and manuscript revision.
Competing interests: The authors declare that they have no conflict of interest.

## Ethical statement

The data used in this study was collected on a voluntary and anonymous basis. Identification of
individual participants in the questionnaire is impossible. Ethics approval was obtained through the
University of Florida's Institutional Review Board (IRB).

## Acknowledgements

We thank the Streaming Science project for providing website hosting and the list-serve for participant
recruitment. We thank Dr. Megan Borel and Laura Mulrooney from the University of Florida for their
help during the field production and recording of the videos. Also, we would like to thank Dr. Anita
Marshall and the Library of Inclusive Field Technology for providing the technical support and
recording devices. Finally, thanks to all participant teachers/classrooms/students for engaging in the
program and helping us collect the required information for this project. We appreciate the enlightening
reviews by Edward McGowan and Janine Krippner which improved this manuscript, and the help
provided by the Geoscience Communication editors. Proofreading and grammar correction of the
manuscript was done using DeepL writing tool.

## Financial Support

This study was supported by a research grant provided by the Florida Chapter of the Association of
Women Geologists, and the Department of Geological Sciences at the University of Florida.

## Data Availability

The authors confirm that the data supporting the findings of this study are available within the article
and its supplementary materials.

Click here to enter text.

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

Click here to enter text.