# Peer review of "Rocks Really Rock: Electronic field trips via Web Google-Earth can"

_EGUsphere, 2023_

## Author Response (AR1)

Answers Reviewer #1

Dear authors, I would first like to start off by saying I enjoyed reading this paper and well done on writing the submission. As a geologist interested in education and scicomm, I am well aware of the situation regarding the decline in student uptake in the subject. Like yourselves, I thoroughly agree the answer lies within promoting the science to younger audiences.

Dear reviewer #1. Thank you for this comment, we are very grateful for your review and interest in these topics. We have received your feedback and have implemented the suggested changes.

Regarding your submission, I have a few questions/suggestions to improve the quality of the manuscript.

Have you read Rogers et al., 2023? – "you just look at rocks, and have beards" Perception of geology from the UK: a qualitative analysis from an online survey.

We were not aware of this publication, but we found its findings insightful and pertinent, and therefore we have added it to our references, and included these authors work through all our manuscript

Lines [108, 472, 509]

As the title suggests, the preprint provides an insight into the perception of geology from prospective students in the UK.

Expanding on the previous comment, you do briefly highlight international associations are aware of the problem ES is facing, but then spend the rest of the manuscript discussing things from a US perspective. This is indeed an international problem – I recently heard Australia has a major recruitment deficit in the mining industry. You should continue to mention the problem is an international one throughout the manuscript, particularly in the conclusion, and how your research can have a global reach beyond US K-12 students.

Thanks for this suggestion, and for pointing out the Australian case. We have acknowledged further the discussion the international- nature of this subject, and included extra citations for this.

Lines [96:99, 103:109, 592:598]

Could you please provide age-ranges for the students who took part in the study so international audiences are able to understand what stage of education K-5 to 12 relates to

in their education system. This only has to be mentioned once in the introduction, then you can continue to use K-12 as you do.

Thank for this suggestion, we have added the explanation of K-12, and the age-range this covers in the introductory paragraph.

The following comments are related to typos/minor comments.

Line 101 – delete one bracket '(' at "learning ((Fitzakerley…"

Done

Line 113 – Perhaps provide some examples of outreach methods used in ES. Then you can say you have opted to focus on using EFTs.

Thank you for this recommendation. We have expanded the last paragraph of section 2.1 to refer to some examples of the diverse avenues and formats of science outreach.

Lines [155:162]

Line 119 – Is '1F1F' meant to be here? If so, please could you expand on what it means.

No, thank you. We have deleted this

Line 193 – lowercase g for 'Geology careers'

done

Line 204 – Typeset heading to be a subtitle.

Done

Line 208 – Is '1' at the end of the sentence a typo?

Yes

Lines 212-220 – Some of these limitations are not readdressed in Section 5.3. It would be beneficial to address some of them in Section 5.3, such as audience fatigue using only one presenter.

Thank you for this great recommendation. We consider this is pertinent, and have added a commentary about this in our recommendations section (Section 5.3)

Lines 259 – sentence states it is talking about attitudes toward geology careers, not geology literacy. It is therefore a direct repetition of Line 248.

Thank you for spotting this. We have corrected the wording accordingly.

Line 263 – delete bracket ')' at "the showed the greatest) change"

Thank you for spotting this. We have corrected the wording accordingly.

I hope this feedback is useful and I wish you all the best moving forward with this project.

Thank you
* * *
Answers Reviewer #2

Dear Carolina and Jamie,

Great work with this manuscript and all of your work done to not only use this new method of reaching students, but also on getting your experience into this format alongside the student surveys so that others can follow in your footsteps.

Dear Reviewer #2. Thank you for this comment. We are very grateful for your feedback.

I have a few queries and suggestions that I hope will add to the clarity of your work.

Do you have any indication of how many additional students, teachers, or classrooms have engaged with your ETF outside of this study? I hadn't heard of the Streaming Science platform before, it would be nice to have a short introduction that explains their reach, like how many users participate through this platform?

I am not based in the USA, is this platform available globally? I see Edward has also suggested providing age ranges for K-12 for an international audience as this is relevant outside of the USA.

Yes, the platform is globally available. We have added that clarification

Line [182]

Thank you. We have provided the age significance of K-12 and age range for the international audience in the introductory paragraph.

Did the teachers who took part have any recommendations or feedback on what they saw really worked? It would be interesting seeing their perspective if that was communicated.

Yes, the 6 surveyed teachers answered a short survey and left feedback to us. We have added this information in Table#7 and Table#8 and discussed this throughout the text.

It would be great to have a short example of how you incorporated storytelling with the content. As a science communicator I understand how important storytelling is for engaging people with our work, to have an example of how this can be done with these topics might help the readers of your manuscript to envision how they could go about creating their own content.

Thank you for this suggestion. We have added to the section 3.1 "EFT context and content development", a section explaining how the storytelling was used to script the content of the EFT videos. In addition, we have also modified section 5.1.3 to make reference to the added content.

Thank you for your work in this area, it's good to see novel ideas on how we can engage students appearing in geoscience literature. I look forward to seeing the finished product.

Thanks so much for your feedback. We are very thankful for your thoughtful suggestions.